# Deciphering the Complex Immunopathogenesis of Alopecia Areata

**DOI:** 10.3390/ijms25115652

**Published:** 2024-05-22

**Authors:** Ingrid Šutić Udović, Nika Hlača, Larisa Prpić Massari, Ines Brajac, Marija Kaštelan, Marijana Vičić

**Affiliations:** Department of Dermatovenereology, Clinical Hospital Centre Rijeka, Medical Faculty, University of Rijeka, Krešimirova 42, 51000 Rijeka, Croatia; ingrid.sutic@medri.uniri.hr (I.Š.U.); nika.hlaca@medri.uniri.hr (N.H.); ines.brajac@medri.uniri.hr (I.B.); marija.kastelan@medri.uniri.hr (M.K.); marijana.vicic@medri.uniri.hr (M.V.)

**Keywords:** alopecia areata, dendritic cells, aetiology, immunopathogenesis, keratinocytes, macrophages, mast cells, NK cells, T lymphocytes

## Abstract

Alopecia areata (AA) is an autoimmune-mediated disorder in which the proximal hair follicle (HF) attack results in non-scarring partial to total scalp or body hair loss. Despite the growing knowledge about AA, its exact cause still needs to be understood. However, immunity and genetic factors are affirmed to be critical in AA development. While the genome-wide association studies proved the innate and acquired immunity involvement, AA mouse models implicated the IFN-γ- and cytotoxic CD8+ T-cell-mediated immune response as the main drivers of disease pathogenesis. The AA hair loss is caused by T-cell-mediated inflammation in the HF area, disturbing its function and disrupting the hair growth cycle without destroying the follicle. Thus, the loss of HF immune privilege, autoimmune HF destruction mediated by cytotoxic mechanisms, and the upregulation of inflammatory pathways play a crucial role. AA is associated with concurrent systemic and autoimmune disorders such as atopic dermatitis, vitiligo, psoriasis, and thyroiditis. Likewise, the patient’s quality of life (QoL) is significantly impaired by morphologic disfigurement caused by the illness. The patients experience a negative impact on psychological well-being and self-esteem and may be more likely to suffer from psychiatric comorbidities. This manuscript aims to present the latest knowledge on the pathogenesis of AA, which involves genetic, epigenetic, immunological, and environmental factors, with a particular emphasis on immunopathogenesis.

## 1. Introduction

Alopecia areata (AA) is a chronic, inflammatory, immune-mediated disease characterized by non-scarring hair loss in the foremost sharply defined areas [1,2]. AA is a common disorder whose incidence ranges between 0.57 and 3.8% in the general population, while its cumulative lifetime incidence is about 2% among Western countries’ inhabitants [2,3]. The illness affects patients of all races and genders without significant predominance; still, current data show that individuals of non-Caucasian origin are more prone to disease development, while men tend to be diagnosed earlier than women [4,5]. AA can affect patients of any age, but 70–80% of them exhibit symptoms before the age of 40. The highest disease prevalence (10–50%) is reported in children, in particular those with a positive family history of AA [6]. An increasing number of patients visit tertiary dermatology centers due to AA, the second most common form of non-cicatricial alopecia [7].

Even though the exact cause of AA is still unknown, genetic and immunological factors are recognized as critical contributors to disease development [4]. Peribulbar infiltration, a constant histopathological feature of acute and chronic AA, and the improvement of the clinical picture after immunosuppressive therapy prove that inflammation is an essential etiopathogenetic factor [8]. During the hair follicle (HF) growth or anagen phase, immunocytes, including plasmacytoid dendritic cells (pDC), helper (Th) and cytotoxic (Tc) T lymphocytes, and natural killer (NK) cells, infiltrate the lower part of the hair bulb, triggering an autoimmune response that leads to anagen HF collapse that clinically presents as alopecia [9,10,11]. The essential initiators of this inflammatory circuit are autoreactive CD8+ cytotoxic T lymphocytes, which, together with Th1, Th17, and NK cells, produce IFN-γ and mediate the HF function disturbance and inhibition of the hair growth cycle and premature hair loss [9].

Although the scalp HFs are the primary targets, AA can also affect the body hair and nails [12,13]. The prevalence of nail involvement is estimated to be 64.1%, with pitting being the most common presentation [14]. Diffuse scalp involvement is uncommon, and AA typically presents as single or multiple areas of hair loss exhibiting various clinical manifestations such as unilocular, multilocular, total, and universal AA or ophiasis, depending on patch size, localization, and extent [15]. While the disease course is unpredictable, with recurrences or progression possible at any stage, the overall prognosis is favorable since 80% of patients have spontaneous hair regrowth within a year after the first AA incidence [13]. However, the worse outcome of the disease is indicated by a positive family history, childhood onset of AA, nail involvement, atopic dermatitis, and the existence of chronic lesions lasting longer than 12 months [16]. In addition to its impact on hair loss, AA is associated with the development of concurrent and potentially disease-modifying disorders, including atopic diatheses, like allergic rhinitis and particularly atopic dermatitis, appearing in more than 30% of AA patients [17,18]; autoimmune diseases such as vitiligo and psoriasis [19]; hidradenitis suppurativa [20]; thyroid disorders [21]; lupus erythematosus [6], pernicious anemia and celiac disease [22]; irritable bowel syndrome [23]; insulin resistance [24]; migraine [25]; audiologic and ophthalmic abnormalities [26,27,28,29]; and psychiatric comorbidities such as sleep disorders, anxiety, alexithymia, social phobia, and paranoid and major depression disorder [1,30,31,32]. AA disfigures the person’s appearance and significantly impacts self-esteem, sociocultural identity, and patient QoL [2,33]. Therefore, providing psychological support and treating comorbid conditions, alongside primary pharmacological AA therapy, are crucial for reducing patient burden and improving overall well-being [34,35]. As shown by a Glickman study, AA patients have systemic inflammation and dysregulation of immune, cardiovascular, and atherosclerosis biomarkers, suggesting a possible systemic approach [36].

Actual knowledge indicates that AA is a multifactorial illness resulting from the interplay of genetic, immunological, and environmental factors, culminating in characteristic disease pathogenesis [9].

## 2. Genetic and Epigenetic Factors in the Development of AA

A positive family history significantly increases the likelihood of acquiring AA, with up to 48% of patients having relatives with the condition compared to only 2% of them in the general population [37]. The inheritance pattern of AA is complex, showing a concordance of 55% in monozygotic twins and a tenfold increased risk for first-degree relatives of affected individuals, suggesting a genetic predisposition with no clear Mendelian pattern within affected families [6].

Initial studies on candidate genes detected the human leukocyte antigen (HLA) class I and II genes as major risk factors in AA pathogenesis, identifying HLA-DQB1*03, HLA-DQB1*04, HLA-DQB1*16, HLA-C*04-01, and HLA-DR as the most significant risk genotypes [4,38]. These genes are closely linked to the effector functions of CD4+ and CD8+ T cells, playing a significant role in the disease phenotype [13,38]. Further insights into AA pathogenesis were gained through genome-wide association studies (GWAS), which identified 14 associated genomic regions involving the innate and adaptive immune system and hair follicle-related genes [39]. Specific single-nucleotide polymorphisms (SNPs) were identified, including those affecting T-cell activation and proliferation, like cytotoxic T lymphocyte-associated protein 4 (CTLA-4), IL-2 receptor A (IL2RA), IL-2/IL-21 locus, Eos or Ikaros family zinc finger 4 (IKZF4), and IFN-γ-producing NK-cell receptor (NKG2D)-mediated cytotoxicity, such as UL16-binding proteins ligands (ULBP) 3/6 and MICA [39]. ULBP is a novel risk gene cluster on chromosome 6q25.1, which encodes NKG2D activating ligands, i.e., stress-induced molecules that alert immune cells via interaction with the NKG2D receptor. Since the majority of NKG2D+ cells are CD8+ T cells, this finding supports their predominant role in AA [40].

Recent GWAS have expanded the initial findings, identifying additional risk loci associated with AA. A genome-wide meta-analysis by Betz et al. revealed an association with genes such as ACOXL/BCL2L11, GARP, and SH2B3 (LNK)/ATXN2 or the ones related to autophagy/apoptosis, regulatory T cells (Tregs), and Janus kinase (JAK) signaling [38]. In addition, the discoveries of IL-17A/IL-17AR gene polymorphisms indicate a possible influence of the Th17 axis, while the IL-4/13 gene involvement reveals the potential participation of type 2 inflammatory events [4]. In contrast, the genome-wide analysis of copy number variants (CNVs) of candidate genes by Fischer et al. identified duplications in melanin-concentrating hormone receptor 2 (MCHR2) and MCHR2 antisense RNA 1 (MCHR2-AS1), highlighting the involvement of genes affecting pigmentation. This finding may explain the whitish discoloration of newly grown hair that often occurs in AA patients after an acute episode of illness [41]. Variants of other implicated genes, such as syntaxin-17 (STX17) and peroxiredoxin-5 (PRDX5), suggested the potential role of oxidative stress in AA pathogenesis [38,42]. Dysregulation of PRDX5 may enable aberrant cells to survive, presenting damaged self-antigens and promoting autoimmune processes [42]. Besides immune aberrations, keratinization disorders could also be involved in the AA pathogenesis since an altered hair shaft constitutive molecule (i.e., coiled-coil alpha-helical rod protein 1 (CCHCR1) was identified [43].

Epigenetic factors have also been implicated in AA pathogenesis [39]. Researchers found that DNA methylation levels were altered in AA patients, and certain epigenetic modifier genes, such as MBD1, DNMT1, and HDAC2, had aberrant expression. [44]. The miRNA expression profile of the C3H/HeJ mouse model of AA demonstrated significant overexpression of mmu-miR-155 and downregulation of mmu-miR-1, mmu-miR-101a, and mmumiR-705 [45]. AA patients with severe and active disease had miR-185-5p, miR-125b-5p, and miR-186-5p expressed on their microarrays, suggesting they might play an important role in alopecia [39].

## 3. Environmental Factors in the Development of AA

Various environmental triggers have been identified as potential catalysts for the onset or exacerbation of AA. Infections with human papillomavirus (HPV), Epstein–Barr virus (EBV), human immunodeficiency virus (HIV), hepatitis B and C viruses, and SARS-CoV-2 have been documented to precede AA development [6,46,47,48,49]. Vaccinations against hepatitis B or COVID-19 infection and certain drug regimens, including highly active antiretroviral therapy (HAART) or amphetamines, are also described as inducible factors [49,50,51]. Additionally, metabolic deficiencies, such as low serum levels of ferritin, folate, zinc, selenium, and vitamin D, have been associated with AA onset and changes in its course and severity [8,52,53,54,55,56,57].

Lifestyle factors also play a significant role in AA [58,59]. Smoking has been linked to a higher incidence of AA, potentially due to pro-inflammatory cytokine dominance induced by cigarette smoke and the overconcentration of free radicals, which can cause hair follicle (HF) immune privilege (IP) loss [60]. While alcohol consumption may contribute to immune imbalances associated with AA, some studies suggest that moderate intake could alleviate psychological stress and positively impact the disease course [58]. Obesity, known to exacerbate various inflammatory skin conditions like psoriasis or atopic dermatitis, is proposed to increase the risk of AA, possibly through IL-17-mediated inflammation [58,61,62]. On the contrary, specific dietary interventions, such as a gluten-free diet rich in omega-3 fatty acids, have shown promise in reducing the risk and severity of AA [58,63].

Psychological stress is a widely recognized trigger for AA, with around 23% of patients reporting an emotional crisis or major traumatic event preceding the onset or worsening of the disease [30]. The established association of HFs with the hypothalamic–pituitary–adrenal axis, where the HFs are both targets and producers of stress hormones, also confirms emotional stress as a triggering pathogenetic factor of AA [64,65]. A study by Talaei et al. concluded that AA patients, in comparison to the control group, show different temperament-character profiles, especially harm avoidance and reward dependence, while patients with the disease relapse reported more psychiatric symptomatic behaviors [66]. Although mental burden undoubtedly contributes to autoimmune dysregulation, it likely interacts with other mentioned factors, as evidenced by AA cases in newborns and infants, whose manifestation cannot be attributed solely to stress [58,67].

There is evidence that oxidative stress may activate NKG2D ligands, disrupt IP, and promote autoimmunity in AA patients. Reduced levels of molecules with antioxidant properties were additionally found in the blood and affected the skin of patients with AA [39]. AA patients have a general pro-oxidative status due to altered oxidative stress (OS) markers in their peripheral blood and lesional skin. The OS stimulates NKG2D receptor expression in CD8+ cytotoxic T cells and NK cells by enhancing the expression of MHC class I chain-related A (MICA) in HF keratinocytes and leads to destabilization of the HF-IP site through the production of IFN-γ that stimulates JAK pathways [68].

The role of microbial dysbiosis in hair loss has been recently pointed out. Microbiota profiling showed both cutaneous and intestinal microbiota alteration in AA patients compared to healthy controls [69,70,71]. Investigation performed by Won et al. revealed a more diverse scalp microbiota in AA patients with a significant increase in Cutibacterium acnes and a decrease in Staphylococcus epidermidis and Staphylococcus aureus. An increase in the α-diversity of the microbiota of AA lesions has also been detected [72,73]. More precisely, an increase in the Anaerococcus and Neisseria, a decrease in Staphylococcus, and a total absence of the SMB53 genus belonging to the Clostridiaceae family was observed at the epidermal level, whereas a reduction in Candidatus Aquiluna, ACK-M1, and Staphylococcus and expansion in Acinetobacter at the dermal level has been identified [74]. The gut microbiota showed its importance in AA pathogenesis, as well. In the mouse C3H/HeJ model, it was proven that the onset of AA could be impacted by the composition of the diet, especially soybean phytoestrogens, suggesting the microbiome’s function [75]. The overall gut microbial composition in AA patients differed from that of healthy controls, with the relative abundance of Actinobacteria and Candidate division TM7 and Bacteroidetes and Fusobacteria being significantly lower [76]. The two study groups had no significant differences at the α-diversity level [69].

The emerging concept of the skin exposome further underscores the role of environmental exposures throughout life in influencing or modifying skin conditions like AA [77].

## 4. Immunopathogenesis of AA

Although the etiopathogenesis of AA has yet to be fully elucidated, its current understanding includes the genetic factors and various environmental triggers, whose interaction influences the autoreactive cytotoxic T-lymphocyte activation and increased secretion of interferon (IFN)-γ in a predisposed individual. Type 1 inflammatory response leads to the loss of the hair follicle immune privilege, overexpression of MHC class I, and consequent autoimmune assault on HFs [4,8].

The typical immune-privileged (IP) status of the proximal portion of the anagen HF protects against inappropriate immune attack and autoimmune hair loss. HF-IP is achieved by defensive physical factors like abundant extracellular matrix or absence of lymphatic vessels and immunological shielding by NK cell suppression and low major histocompatibility complex (MHC) expression, which form an anergic state [8,78,79]. Downregulation of MHC class I and local immunoinhibitory environment is additionally ensured by secretion of immunosuppressive molecules, known as “IP guardians”, including α-melanocyte-stimulating hormone (α-MSH), transforming growth factor-β1 (TGF-β1), indoleamine-2,3-dioxygenase (IDO), protein red encoded by IK gene (red/IK), interleukin (IL)-10, calcitonin gene-related peptide, insulin-like growth factor-1, and somatostatin [39], while reduction of MHC class II expression on HF Langerhans cells impairs antigen-presenting cell (APC) function [8]. The IP environment additionally prevents inflammation and local tissue damage by achieving reduced expression of MHC I chain-related gene A (MICA) and UL16-binding protein (ULBP), thus suppressing NK cell activation [8].

However, in AA, these defensive mechanisms are facilitated by the upregulation of MHC I/II, adhesion molecules, and NKG2D ligands; increased secretion of cytokines in HF, such as IFN-γ, IL-15, and IL-2; and decreased IP guardians. Accordingly, autoantigen presentation by melanin-generating anagen HF cells is enhanced, leading to the loss of HF-IP, which is held to be a major event in AA pathogenesis [39,46]. The incriminating autoantigens are thought to be part of HFs keratinocytes, melanocytes, and dermal fibroblasts, such as trichohyalin, glycoprotein 100, tyrosinase and tyrosinase-related protein 1/2 (TRP1/2), retinol-binding protein, and melanoma antigen [8,46,80]. Subsequently, to autoantigen presentation, T lymphocytes migrate to the lower part of the hair bulb and release pro-inflammatory mediators IFN-γ and interleukin (IL)-2, promoting the infiltration of additional CD4+ and CD8+ T cells and other immunocytes into the IP zone [81,82]. This inflammatory response causes keratinocyte apoptosis and severe impairment of the hair growth cycle and HF function, finally resulting in dystrophic anagen or telogen effluvium and AA [6,39] (Figure 1).

Namely, under physiological circumstances, the HF undergoes cyclic phases of anagen (active growth), catagen (involution with epithelial cells apoptosis), and telogen (resting), regulated by complex molecular mechanisms and signaling cascades involving HF components, immune cells, dermal fibroblasts, and skin-associated adipocytes [83]. Conversely, in AA, inflammatory cells infiltrate the HF during the catagen phase, exposing HF autoantigens to immunocytes to a greater extent [83]. The heightened inflammation prematurely shifts anagen HFs into the telogen phase, leading to increased hair shedding [83]. During acute disease stages, perifollicular CD4+ and intrafollicular CD8+ T lymphocytes form extensive infiltrates around anagen HFs, described histologically as a ‘swarm of bees’, disrupting matrix cell proliferation and inducing HF keratinocyte apoptosis, ultimately inhibiting cell division within the hair matrix and producing dystrophic hair shafts [13,83]. On the other hand, in chronic AA stages, HFs become miniaturized due to reduced inflammatory cell presence in the peribulbar space [83]. However, it is important to note that the AA can be reversed since the HF’s epithelial stem cells in the bulge area are unaffected [39].

Based on dysregulated genes identified in AA, it is increasingly evident that both innate and adaptive immunity contribute to disease pathogenesis [10,13]. Upregulated genes involved in antigen presentation and co-stimulation point toward the participation of DC and T lymphocytes in the development of antigen-specific immune-mediated pathology. Furthermore, the presence of genes such as NKG2D indicates the involvement of cytotoxic T lymphocytes and NK cells, while the upregulation of immunoglobulin genes in later stages of disease implies the engagement of B cells and antibody production [13,84,85]. Mast cells (MCs) and eosinophils are consistently found in the lesional skin at all stages of the disease. The density of OX40L mast cells is significantly elevated in AA lesional skin, especially HF mesenchyme and perifollicular tissue [46]. A close connection between MCs and CD8+ T lymphocytes has been recently established since MCs secrete molecules such as TNF-α and CCL5 that inflict the T cells’ action, while activated CD8+ T cells enhance the expression of costimulatory molecules on MCs [86]. However, MCs precise role remains elusive [83].

Researchers have demonstrated that cell-mediated immunopathogenesis is implicated in AA based on animal studies where transferring cultured cells isolated from skin-draining lymph nodes of affected C3H/HeJ mice induced disease in healthy counterparts, highlighting the significance of CD8+NKG2D+ T cells [83,87]. Through a positive feedback loop involving IFN-γ production and IL-15 signaling, these T cells perpetuate type I cellular autoimmunity, contributing to disease onset and progression. IL-15 and other pro-inflammatory cytokines activate JAK-STAT signaling, suggesting its role in AA onset and progression. Hair regrowth in C3H/HeJ AA mice promoted by inhibition of JAK-STAT signaling with systemic administration of JAK inhibitors (JAKi) supports this mechanism [83,88].

Finally, autoantibodies to follicular components have been detected in AA patients. Identified candidate autoantigens include the 44/46 kDa hair-specific keratin and trichohyalin, an essential intermediate filament-associated protein. A total of 44/46 kDa hair-specific keratin is expressed in the precortical zone of anagen HFs, while trichohyalin is expressed in the inner root sheath of the growing HF [85]. The study of Tobin et al. showed the up to 7- and 13-times greater serum levels of anti-HF IgG antibodies and five studied antigens in AA patients than in controls [84]. Whether antibodies to HF-specific proteins contribute to the onset and progression of disease or only represent an autoimmune epiphenomenon must be investigated. Either way, antibodies are suggested to play a particular role in AA pathogenesis since the topical immunotherapy that reduces their levels positively affects hair regrowth [13,89]. Still, further investigations are needed to elucidate their exact contribution to disease progression.
Figure 1Immune privilege collapse of the anagen hair follicle in a patient with AA. Healthy HFs’ immune-privileged status is ensured by the lower expression of MHC I/II and APC and upregulation of immunosuppressive molecules, known as “IP guardians”, such as α-MSH, TGF-β1, and IL-10. The breakdown of HF-IP is essential for the onset of AA, and it occurs when the balance of signaling pathways upholding this IP is overwhelmed by those leading to collapse. In AA patients, IP compromises the upregulation of MHC I/II, adhesion molecules, and NKG2D ligands; migration of T lymphocytes in the lower part of the hair bulb; increased secretion of cytokines, such as IFN-γ, TNF-α, IL-2, and IL-15; and decreased IP guardians. Adapted from [90].
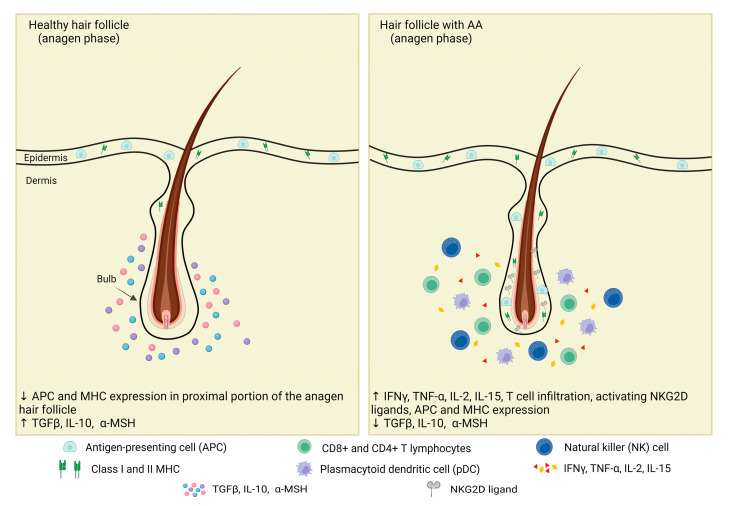


### 4.1. Main Immunocytes Involved in AA Inflammatory Networks

#### 4.1.1. CD8+ Lymphocytes

Cytotoxic CD8+NKG2D+ T lymphocytes are the primary immunocytes that infiltrate the surroundings of HFs and are held to be the key cells that drive the disease pathogenesis (Figure 2). NKG2D is an activating receptor expressed on CD8+ T cells and NK cells which recognize NKG2D ligands, like ULBP3/6 and MICA, and then upregulate MHC expression, which is crucial in mediating HP-IP collapse [8]. CD8+ T cells mainly infiltrate the intrafollicular area, and their denser infiltrations correlate with greater histopathology of AA-affected hair bulbar regions [91]. The clinical significance of CD8+ lymphocytes in AA is underscored by the correlation between CD8+ T-cell density and disease severity [5].

Animal studies have demonstrated that CD8+ T lymphocytes are necessary and sufficient for inducing the disease [92,93,94]. Transfer of these cells to C3H/HeJ mice can initiate AA, while depletion of CD4+ and CD8+ T cells by using monoclonal antibodies has been shown to promote hair regrowth in AA-affected mice [5]. Additionally, upon engraftment of AA skin on mice, transcripts of the CD8a, encoding the α component of the CD8 costimulatory molecule on CD8+ T cells, steadily increase, suggesting the early recruitment of CD8+ T cells in the disease course [95]. In HFs of AA patients, upregulation of the NKG2D encoding gene and overexpression of its associated ligands NKD2DL3 and RAET1L have been detected, with levels significantly elevated compared to controls or patients with other inflammatory scalp diseases [40]. Transcriptional profiling of both mouse and human-affected skin has revealed cytokine production indicative of Tc infiltration, including increased levels of IFN-γ and γ-chain (γc) cytokines and receptors, known to promote the activation and survival of IFN-γ–producing CD8+NKG2D+ effector T lymphocytes [13,88].

Similar to other autoimmune dermatoses, like psoriasis and lichen planus, the previous research revealed that cell-mediated cytotoxicity and apoptotic mechanisms involving cytotoxic molecules, such as granzyme B, granulysin, and perforin, and the Fas–Fas ligand pathways form a central component of AA pathogenesis [96,97,98,99,100,101,102]. It has been shown that effector CD8+ T cells and NK cells use serine protease granzyme B (GZMB) to target HFs. After releasing cytotoxic granules, GZMB enters the target cell with the assistance of pore-forming protein perforin and finally causes cell apoptosis. Besides its cytotoxicity, GZMB has a proteolytic activity, along with augmentation of cytokine/growth factor activity [103]. Elevated levels of GZMB and cytotoxic granule-associated RNA-binding protein I (TIA1) have been observed in human HFs affected by AA. At the same time, only GZMB transcripts were increased in AA-affected C3H/HeJ mice, suggesting the essential role of GZMB secretion in CD8+ T-cell-driven pathology [13]. GZMB has been found within the primary site of immune collapse, i.e., HF isthmus, and was proposed as a downstream mediator of the JAK/STAT pathway [103]. Koguchi-Yoshioka et al. recently confirmed augmented GZMB production in the lesional skin of AA patients. Moreover, they found significant CD49 overexpression and enlarged GZMB production in the lesional CD8+ T cells in nonresponding patients compared to responders, indicating that CD49a+ CD8+ GZMB-producing T cells in the lesional AA skin may reflect the disease prognosis [104]. The mentioned results directly predict disease severity and affect the therapeutic options since GZMB expression is strictly regulated by JAK, and JAKi reduces the GZMB expression in the skin of treated patients. These findings allude that the topical use of drugs that reduce GZMB expression could be a possible and safer therapeutic option in the future [104].

Another proapoptotic mediator released from activated Tc cells is granulysin (GNLY). Ono et al. proved the significant elevation of serum GNLY in both acute and chronic AA patients, where GNLY levels were associated with broader bald skin areas, poorer prognosis in acute AA cases, and the co-existence of allergic disorders in AA patients. The affected tissue analyses showed the presence of perifollicular GNLY-bearing cells consistent with dense CD8+ T-cell lymphocytic infiltration and were mainly detected in acute AA lesions. Therefore, the study concluded that GNLY represented cell-mediated cytotoxicity, while its serum levels were proposed as markers for the disease activity in the acute phase of AA [105]. The study of Oba et al., in turn, suggested GNLY as a potential mediator of HF attack and concluded that the GNLY serum level could be a good correlate of AA immunological activity and an indicator of the JAKi therapeutic effectiveness [106].

#### 4.1.2. CD4+ Lymphocytes

CD4+ T cells are found alongside CD8+ T cells in the HFs during the anagen phase in AA-affected skin [95]. Primarily, two subtypes of CD4+ T cells contribute to autoimmunity in AA, i.e., T helper 17 (Th17) cells and regulatory T (Treg) cells [107,108]. Th17 cells, characterized by the production of IL-17, IL-22, and IL-23, infiltrate the dermal area in the proximal vicinity of HFs in AA patients, taking the helper role together with Th1 to induce inflammation and contributing to cell-mediated autoimmunity [108,109]. Additionally, Th17 cells are implicated in autoimmunity processes in psoriasis and vitiligo, two autoimmune diseases often comorbid with AA [108,110]. CD4+ T cells mainly infiltrate the peribulbar area in the dermis [91]. A study by Hong et al. found an admix pattern of Th1 and Th17 cells in the bulb and bulge area of HFs, where the dense infiltrations of CCR6+ Th17 cells were associated with extensive stem cell destruction and more severe hair loss in patients with chronic AA [91].

Tregs, a specialized subpopulation of T cells, suppress the immune response, maintaining homeostasis and self-tolerance by cell-to-cell contact or secreting soluble factors such as TGF-β [111]. Abnormal ratios of Th17 and Treg cells have been observed in AA patients where, during active stages of the disease, Th17 levels exceed Treg levels in the blood. In contrast, in more severe AA, levels of circulating Tregs surpass Th17 levels [107]. Decreased serum levels of regulatory cytokine TGF-β were found in patients with active AA supporting the altered function of Fox3+ Tregs [111,112]. The imbalance between Th17 and Treg cells in AA patients leads to inflammation and autoimmunity through pro-inflammatory mechanisms similar to those reported in other autoimmune diseases [113]. In a murine model of chronic AA, monoclonal antibody depletion of CD4+ and CD8+ T cells significantly improved hair regrowth [13]. Recent studies have shed light on the role of resident memory T cells (Trm) in inflammatory skin diseases, such as psoriasis or vitiligo. These studies have also shown that GZMB-expressing CD49a+ CD8 T cells may serve as Trm in AA [104]. This new understanding of Trm’s involvement in AA adds to our knowledge of the disease’s pathogenesis. Overall, increased T-cell immunity, coupled with a breakdown of immune tolerance because of Treg deficiency, facilitates AA.

#### 4.1.3. Plasmacytoid Dendritic Cells

In addition to CD8+ and CD4+ T lymphocytes, plasmacytoid dendritic cells (pDCs) are also present in the infiltrates around HFs of AA patients [11]. The pDCs are specialized dendritic cell populations with plasma cell morphology that bridge the innate and adaptive immune responses by modulating the function of myeloid dendritic cells (mDCs), NK cells, and T and B lymphocytes [13,114]. They contribute to the disease by activating, differentiating, and producing cytokines and chemokines by T lymphocytes, thereby amplifying the inflammatory process. While pDCs are typically absent in normal skin, they can infiltrate the skin and contribute to inflammation or autoimmunity in conditions such as lupus, psoriasis, or AA following injury or pathological processes [11,114,115]. Upon activation via Toll-like receptors, pDCs produce large quantities of type I interferons, such as IFN-α and β, which in turn induce responses by CD4+, CD8+, and NK cells toward the HFs in AA patients [114]. Sato-Kawamura et al. found the presence of APC, such as Langerhans cells and dendritic cells, in the inflammatory infiltrate of lesional AA skin. The perifollicular and follicular APC, expressing CD1a, CD40, CD54, and HLA-DR, showed interaction with infiltrating T cells to produce IFN-γ, thus depriving dermal papilla cells of their ability to maintain anagen hair growth [116]. Although the exact mechanism of pDC recruitment in the HFs area is not fully understood, their function is undoubtedly substantial, as they act as a link between the innate and adaptive immune responses, ultimately leading to hair loss in AA [11,13].

#### 4.1.4. NK Cells

The primary function of NK cells upon their activation is eliminating infected and damaged cells through cytotoxic mechanisms, along with cytokine secretion, including IFN-γ, TNF-α, and TGF-β. HFs are well-known immune-privileged sites that can evade autoimmune reactions by enhancing suppressive signaling around them, thereby impairing the function of CD8+ cells and NK cells. However, when this IP is disrupted, immune cells, including NK cells (CD56+ NKG2D+), infiltrate the lower part of HF [117]. Since pDCs are the main sources of type I IFNs in AA, they enhance the activity of NK cells and CD8+ and CD4+ cells [13]. Once NK cells and CD8+, Th1, and Th17 cells are activated, they produce large quantities of IFN-γ [10]. IFN-γ disrupts the hair growth cycle by inducing JAK/STAT signaling. In healthy individuals during the anagen phase, JAK/STAT signaling is suppressed because its activation inhibits the proliferation and activation of hair stem cells [118,119]. Therefore, IFN-γ induced JAK/STAT signaling could be the reason for the premature termination of the anagen phase in AA [120]. While NK cells undoubtedly play a role in AA pathogenesis by producing IFN-γ, further research is needed to enlighten their importance fully.

### 4.2. Main Cytokines Involved in AA Inflammatory Networks

Numerous mediators contribute to initiating and perpetuating inflammation in AA, with one of the central cytokines being IFN-γ, the principal mediator of type 1 inflammation [112]. Within AA skin lesions, IFN-γ is primarily produced by CD8+ T cells and autoreactive Th1, Th17, and NK cells. IFN-γ disrupts the immune privilege of HFs, which is typically maintained during the anagen phase through downregulation of MHC class I molecules and upregulation of inhibitors of NK and CD8+ cells, such as macrophage migration inhibitory factor (MIF) and transforming growth factors (TGF) β1 and β2 [112,120]. IFN-γ triggers the collapse of HF immune privilege by inducing ectopic expression of MHC-I molecules, activating ligands for NK-cell receptors (NKG2D), or stimulating chemokine secretion, such as CXCLs [112].

IFN-γ levels are upregulated in AA patients’ serum and lesional skin, correlating with more severe clinical features of AA, especially with widespread AA, alopecia totalis, or universalis [91,112]. The critical role of IFN-γ in AA pathogenesis is supported by the studies with IFN-γ gene knocked out C3H/HeJ mice, which display resistance to disease induction [78]. Additionally, antibodies targeting IFN-γ have effectively prevented hair loss in animal models [78]. During early AA development, increased expression of IFN-γ-induced chemokines such as CXCR3, CXCL-9, CXCL-10, and CXCL-11 is observed, facilitating immune cell recruitment and amplifying Th1- and NK cell-mediated responses [78]. The recruitment of lymphocytes at the HF can result in the onset of AA. In contrast, the positive IFN-γ feedback loop can explain the duration and progression of the disease by maintaining the lymphocytic infiltrates and enhancing Th1 activities [121]. CXCR3 is expressed primarily on Th1 CD4+ T cells, CD8+ T cells, and NK and NKT cells, while CXCR3 ligands (CXCL-9, CXCL-10, and CXCL-11) are secreted by many tissue-resident cells including dendritic cells [13]. By CXCR3 blockade, the inhibition of the downstream IFN-γ signaling and prevention of the AA development in C3H/HeJ mice can be achieved by inhibiting pathogenic T-cell recruitment [122]. Accordingly, the blockade of CXCR3 may be a potential target for future therapeutics [5].

Furthermore, IFN-γ disturbs the HF function and the hair growth cycle by inducing JAK/STAT signaling, resulting in premature hair loss and hair growth inhibition without follicle destruction [4,90]. JAK/STAT signaling, typically suppressed during the anagen phase in healthy individuals, inhibits the proliferation and activation of hair stem cells [119]. Inhibition of the JAK/STAT signaling pathway has shown promise in reversing symptoms of AA and promoting hair regrowth [119].

Another vital cytokine implicated in AA immunopathogenesis is tumor necrosis factor-alpha (TNF-α). TNF-α, a pro-inflammatory cytokine produced by the infiltrating T cells in AA skin, exerts both damaging and protective effects in autoimmune disorders [112]. Increased TNF-α levels are found in AA patients’ serum, and its concentration correlated with female gender, AA severity, and illness duration longer than six months [112,123]. Studies on ex vivo HFs show that TNF-α interferes with keratinocyte differentiation and causes hair cycle disturbance, inducing the catagen phase. Despite its anti-proliferative effects on epithelial cells and keratinocytes, blocking TNF-α has been largely ineffective in AA treatment and may even exacerbate the disease in some cases [124,125]. Therefore, elevated TNF-α levels in AA patients may protect against IFN-α and IFN-γ responses, inhibiting MHC class I upregulation caused by IFN-γ and suppressing the development of IFN-α producing pDCs, responsible for CD4+, CD8+, and NK cell action [13]. The antagonism of TNF-α probably leads to AA by allowing uncontrolled IFN-α production from pDCs and interfering with the protection cytokine could provide against IFN-γ upregulation of MHC class I [13]. IFN-γ and TNF-α are the first cytokines produced and released around HFs [78].

Additional important cytokines in AA development are structurally similar to IL-2 and IL-15, which signal through two shared receptor subunits, IL-2/15 β and γ chain, and whose production is stimulated by IFN-γ overexpression [112]. Increased expressions of IL-2 and IL-15 and their receptors on CD8+ T cells around the HFs, as well as in the serum of AA patients, have been observed [112]. Blocking these interleukins or their receptors prevents AA in mice by reducing the accumulation of CD8+NKG2D+ T cells and dermal interferon responses [78,126]. IL-2 and IL-15 are produced by activated immune cells or stressed keratinocytes and exert effects such as limiting the suppressive effect of Tregs, promoting NKG2D expression on NK cells, and activating the NKG2D signaling pathway leading to Janus kinase activation and shifting CD8+ T cells toward cytotoxic functions independent of TCR [127,128,129]. Activated cytotoxic lymphocyte T cells stimulated by IL-15 trigger additional IFN-γ secretion through the JAK-STAT pathway, forming an inflammatory circuit accountable for the maintenance phase of AA [88].

IL-12, a pro-inflammatory cytokine produced by DCs and macrophages, is another key player in AA. It induces IFN-γ production by T cells and NK cells, and its serum levels were found to be higher in AA patients [112]. A study conducted by Rossi et al. revealed a correlation between serum IL-12 levels and the grade and duration of AA [123].

Several studies have indicated increased levels of IL-17 and IL-22 in patients’ lesional skin and serum, revealing the influence of the Th17 pathway in AA inflammation [130]. The level of IL-17 is particularly elevated in the immediate environment of HF [108]. IL-17 showed a positive correlation with early onset and severity of AA, while its concentration significantly decreases after AA therapy, such as phototherapy, diphenylcyclopropenone, or JAK inhibitors [109]. Serum level of IL-22 has been positively correlated with AA duration and depression [112]. Limited data considering IL-23 serum level also show its upregulation [112].

Alternatifvely to Th1 and Th17 inflammatory routes, recent research has proven the excessive expression of type 2 molecules, such as cytokines IL-4/-6/-10/-13/-31 and -33 and chemokines CCL17 and CCL18, in AA lesions and serum of patients, indicating their contribution to the pathogenesis of the disease [126,131].

Although AA is traditionally considered a Th1-mediated disease, current knowledge based on animal models and preclinical and translational studies imposes immunological heterogeneity, with a tight interplay of Th1, Th17, and Th2 responses [132]. The previous data support the thesis that AA is characterized by the dysregulation of systemic Th1, Th2, and Th17 cytokines, indicating that AA is a systemic inflammatory disorder whereby patients with longer duration of AA and its more severe forms are more exposed to the harmful effects of systemic inflammation [112].

## 5. Current Treatment and Future Perspectives

A new understanding of the immunopathogenesis of AA has led to treatments that significantly improve the patients’ QoL. In the US and Europe, two oral systemic therapies are currently approved for the treatment of AA. JAK 1/2 inhibitor baricitinib and selective JAK3/TEC (tyrosine kinase expressed in hepatocellular carcinoma) kinase inhibitor ritlecitinib. Both baricitinib and ritlecitinib are approved for the treatment of severe AA in adults, while ritlecitinib is also safe to treat severe AA in adolescents. Several JAK inhibitors with various profiles of selectivity toward JAK isoforms, such as deuruxolitinib, jaktinib, ivarmacitinib, KL130008, and deucravacitinib, are under investigation for the treatment of several immunoinflammatory diseases, including AA [133]. Other traditional systemic medications used in AA include glucocorticoids, cyclosporine, methotrexate, and other anti-inflammatory drugs or contact immunotherapy for immunomodulation [134].

With an increased understanding of AA’s T-cell-mediated autoimmune and inflammatory pathogenesis, additional therapeutic pathways controlling T-cell recruitment in HF and upstream cytokine signaling are being explored to develop AA therapies. Other T-cell-related targets of interest for future treatment development include Tregs, immune tolerance, and the microbiome [133].

Given the complex multifactorial nature of the signaling pathways underlying the immune-mediated attack on HF in AA, there is a requisite for developing therapies that target multiple distinct signaling pathways. By using dual-targeted therapies, we could overcome the ceiling effect of current therapies, such as immune escape and drug resistance [133,134].

## 6. Conclusions

AA is a common tissue-specific autoimmune disease characterized by localized nonscarring hair loss on the scalp or other hair-bearing areas. This condition involves a multifaceted interplay of genetic, immunological, and environmental factors that trigger immune alterations, cause the imbalance between factors preserving the HF-IP, and lead to premature hair loss. The central effectors of AA immunopathogenesis are IFN-γ-proapoptotic molecule-secreting CD8+NKG2D+ T cells, which, together with CD4+ T cells, plasmacytoid dendritic cells, and NK cells, orchestrate the inflammatory response within hair follicles, disrupting the hair growth cycle, inducing JAK/STAT signaling, and leading to premature hair loss. While traditionally viewed as a type 1 inflammatory disease, recent research highlights dysregulation in Th2 and Th17 pathways, as well as the influence of epigenetic factors, the microbiome, nutritional deficiencies, and exposome. Understanding these complexities offers insights into potential therapeutic targets. Current treatments, such as JAK inhibitors, show promise in managing AA, and ongoing research into cytokine modulation and cellular interventions holds potential for future treatments. In conclusion, unraveling the intricate immunopathogenesis of AA expands our comprehension of this puzzling condition and opens avenues for innovative therapeutic interventions. Further investigation into its underlying mechanism promises to enhance our ability to manage the disease effectively, particularly in severe cases, ultimately improving the QoL for affected individuals.

## Figures and Tables

**Figure 2 ijms-25-05652-f002:**
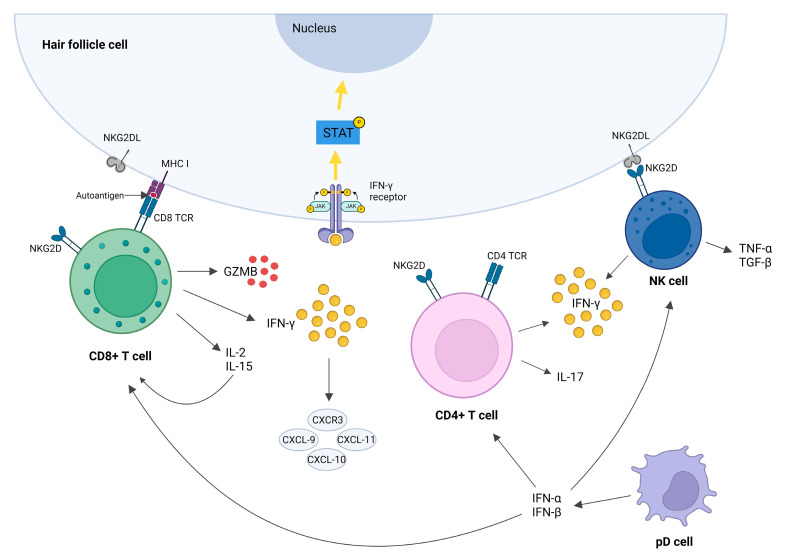
Immune cells responsible for AA development. Cytotoxic CD8+ T cells (CD8^+^NKG2D^+^ T cells) are the first immune cells that infiltrate the HF surrounding and are considered to be the main drivers of disease pathogenesis. CD8^+^NKG2D^+^ T cells produce IL-2, IL-15, GZMB, and IFN-γ. IL-2 and IL-15 maintain CD8+NKG2D+ T cells via a positive feedback loop. GZMB promotes cell lysis, while IFN-γ triggers collapse of HF-IP by inducing JAK/STAT signaling. NK cells also express the NKG2D and may attack the HF cells upon binding the NKG2D ligand in similar manner as the CD8+ cells. Effector CD4+ and NK cells both produce IFN-γ, which then further induces the production of chemokines (CXCR3, CXCL-9, CXCL-10, and CXCL-11) and maintains inflammation by attracting other immune cells into the peribulbar space. Plasmacytoid dendritic cells (pDCs) produce large quantities of type I interferons (IFN-α and β), which induce CD4+, CD8+, and NK responses toward the HFs in AA patients. Adapted from [9,13].

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
