# Peer review of "Deciphering the Complex Immunopathogenesis of Alopecia Areata"

_ijms, 2024, doi:10.3390/ijms25115652_

Round 1

Reviewer 1 Report

Comments and Suggestions for Authors

In this manuscript, the authors aim to review the immunopathogenesis of Alopecia areata (AA), an autoimmune-mediated disorder in which the proximal hair follicle attack results in non-scarring partial to total scalp or body hair loss. The authors first discussed the genetic, epigenetic, and environmental factors in the development of AA, followed by immunopathogenesis of AA and conclusion. I have the following points for the authors to improve their review manuscript.

1.     For Figure 1, the authors should provide more explanation of the figure in the figure legend (similar to Figure 2)

2.     It would be better if the authors could include another section discussing the potential treatments of AA after the section “Immunopathogenesis of alopecia areata.” This will increase the clinical relevance and readership of this review manuscript. In this new section, the authors can discuss both treatments/cures available clinically and some potential treatments/cures under experimentation. A table of the treatments and associated mechanisms can also be provided, with additional notes from previous sections.

3.     Some English language usage is not organized and with minor mistakes. For example, Alopecia areata has been abbreviated earlier as AA, but line129, line174, line188 and more lines continue to use the full term. The authors should correct this as well as other similar mistakes. Language proofreading by a colleague is recommended.

Comments on the Quality of English Language

English is fine overall. 

Author Response

Honoured reviewer,

We are very thankful for your constructive remarks and suggestions.

We have taken all suggestions into account and accordingly:

- used the abbreviation AA throughout the text 

- wrote a description of the image below in Figure 1

- made specific corrections to the English language

We sincerely thank you for the suggestion about adding a section on therapy. In this article, we aimed to present current knowledge about the immunopathogenesis of AA. At the same time, we intend to present a more detailed elaboration of data on current therapy and therapy in development in the following separate article. Respecting your recommendations and the wish to maintain the primary goal of our text, we inserted a short, new section on current and emerging therapies at the end of the text.

Best regards,

Larisa Prpic Massari

Reviewer 2 Report

Comments and Suggestions for Authors

The review article entitled Deciphering the Complex Immunopathogenesis of Alopecia Areata by Ingrid Šutić Udović et al.

The article has not been written professionally neither the aim of this study is clear. I would recommend the rejection of this article.

The authors should extensively revise this article and resubmit it.

Abstract

What was the aim of this study?

Introduction

Extensive revision is needed.

Figures quality is very bad.

Add a paragraph on Limitations.

And add another paragraph on future recommendations.

Conclusion

Should be revised,

Comments on the Quality of English Language

The review article entitled Deciphering the Complex Immunopathogenesis of Alopecia Areata by Ingrid Šutić Udović et al.

The article has not been written professionally neither the aim of this study is clear. I would recommend the rejection of this article.

The authors should extensively revise this article and resubmit it.

Abstract

What was the aim of this study?

Introduction

Extensive revision is needed.

Figures quality is very bad.

Add a paragraph on Limitations.

And add another paragraph on future recommendations.

Conclusion

Should be revised,

Author Response

Honoured reviewer,

We are very thankful for your remarks and suggestions for our review article, which we have appreciated and, accordingly:

- added the aim of the review article

- made the corrections of the text and English language

- changed the previous pictures with new ones made in the graphic program BioRender

- added the description of Figure 1

- added a new section related to current and emerging therapy

Best regards,

Larisa Prpic Massari

Round 2

Reviewer 1 Report

Comments and Suggestions for Authors

This review has been revised by the authors and its quality is improved.  

Comments on the Quality of English Language

English is fine. 

Reviewer 2 Report

Comments and Suggestions for Authors

The authors have revised the manuscript and can be accepted for publication now.